# Simulation-Based Approach for Lookahead Scheduling of Onshore Wind Projects Subject to Weather Risk

Emad Mohamed [1], Parinaz Jafari [1], Adam Chehouri [2] and Simaan AbouRizk [1,*]

1 Department of Civil and Environmental Engineering, University of Alberta, Edmonton, AB T6G 2W2, Canada; ehmohame@ualberta.ca (E.M.); parinaz@ualberta.ca (P.J.)
2 DNV Energy Systems Canada Inc., Montreal, QC H1Y 3N1, Canada; adam.chehouri@dnv.com
* Correspondence: abourizk@ualberta.ca

**Abstract:** Executed outdoors in high-wind areas, adverse weather conditions represent a significant risk to onshore wind farm construction activities. While methods for considering historical weather data during pre-construction scheduling are available, approaches capable of quantitatively assessing how short-term weather fluctuations may impact upcoming construction activities have yet to be developed. This study is proposing a hybrid simulation-based approach that uses short-term precipitation, wind speed, and temperature forecasts together with planned and as-built activity durations to develop lookahead (e.g., upcoming 14-day) schedules for improved project planning and control. Functionality and applicability of the method was demonstrated on a case study of a 40 MW onshore wind project, and the method was validated using event validity, face validation, and sensitivity analysis. As expected, favorable weather conditions experienced during the tested lookahead periods resulted in a negligible impact (less than 10% reduction) on the productivity of weather-sensitive activities, which translated into a project delay of one day. The responsiveness of the framework was confirmed through sensitivity analysis, which demonstrated a 50% reduction in productivity resulting from poor weather conditions. The ability of the method to provide decision-support not currently offered by commercially-available scheduling systems was confirmed by subject experts, who endorsed the ability of the method to enhance lookahead scheduling and to facilitate the monitoring and control of weather impact uncertainty on project durations.

**Keywords:** renewable energy; onshore; wind farm; construction; risk; uncertainty; weather; lookahead; simulation

## 1. Introduction

Many countries are transitioning their energy production towards renewable sources to reduce greenhouse gas emissions and meet sustainability targets. Since 2009, global onshore wind capacity has increased fourfold, producing more than 594 GW in 2019 alone [1]. To meet the growing demand, installation of additional wind farms and supporting infrastructure is necessary. However, onshore wind projects are constructed in outdoor environments characterized by high wind speeds [2] and other adverse weather events that can result in significant construction delays [2–4]. Indeed, the schedule delay of an average wind farm project was estimated to be approximately 10% of the planned project duration [5]. Delays in wind farm construction are particularly problematic [6], as most contracts between the owner and the engineering, procurement, and construction (EPC) contractor(s) include a provision for liquidated damages if a project exceeds the contractually-specified end date [7]. Completing wind farm construction tasks on time is essential for ensuring profitability [6].

Variability is an inherent characteristic of construction projects, with as-built progress regularly deviating from planned schedules. To mitigate delays, practitioners often schedule projects from two perspectives, generating (1) a master schedule that provides a holistic view of the entire project and (2) a lookahead schedule that provides a short-term, detailed

work plan for upcoming tasks [8]. Together, master scheduling and short-term or lookahead scheduling are key elements of successful project delivery [9]. Master scheduling provides a global view of a project that can be used for long-term coordination, rough budgeting, and bid preparation [9–11], while lookahead scheduling is used for ongoing performance analysis, increasing reliability of detailed work plans, and for identifying and implementing effective corrective actions during execution [9,10,12].

The evaluation of the impact of weather on construction project scheduling has been addressed and successfully applied by numerous research studies [2,13–17]. These approaches use historical weather data (either extreme conditions or averages) to predict the weather-related impact on project durations when generating master project schedules. Weather can have either a negative or positive impact on productivity and, subsequently, on project duration. When weather conditions are better than expected, productivity of construction projects can improve and schedules may be shortened, representing an opportunity for the project team. In contrast, when weather conditions are worse than expected, productivity of construction projects may be reduced, representing a threat to project success. Although useful for high-level pre-construction scheduling and bid preparation, the suitability of these approaches for short-term lookahead scheduling is primarily limited by two factors. First, as a consequence of being designed for long-term master scheduling, existing approaches are not able to incorporate as-built data, limiting the reliability of lookahead schedules. Second, previous methods use average or extreme weather data that do not capture fluctuations in weather conditions that occur during project execution [18]. These limitations have resulted in the development of unrealistic lookahead schedules, preventing practitioners from identifying and implementing corrective actions in the timeframe required to mitigate weather-related delays. Indeed, a model capable of integrating as-built information, while simultaneously considering short-term weather forecasts to improve lookahead scheduling, has yet to be described in the literature.

To address the aforementioned gaps, this research is proposing a simulation-based approach that integrates as-built information with short-term weather forecast data to improve lookahead scheduling in onshore wind farm construction. The proposed framework includes a newly-developed and generic hybrid simulation model (i.e., discrete-event and continuous) of both weather-sensitive and non-sensitive activities in onshore wind farm construction. Short-term weather precipitation, wind speed, and temperature data are used to derive a productivity factor for weather-sensitive activities. These data—together with as-built information—are input into the model, which generates a 14-day lookahead schedule. Functionality and validity were demonstrated following the application of the proposed framework to a case study of a real onshore wind farm construction project. As the first reported framework to integrate as-built construction information and short-term weather forecast data for lookahead scheduling, the practical application of this approach is expected to improve the evaluation, understanding, and monitoring of weather uncertainties on project execution for improved project management and control of wind farm construction.

## 2. Literature Review

### 2.1. Adverse Weather in Construction Projects

Adverse weather was ranked as the second leading cause of construction-related claims in Canada [19] and as the third most critical risk factor for construction schedule overruns in Jordan [20]. The negative consequences of adverse weather on construction projects includes reduced productivity, work stoppage, and ruined materials, which often result in schedule delays, cost overruns [21–23], and disputes between project stakeholders [18,24]. Weather-related parameters that were extensively studied in construction include precipitation, air temperature, and wind speed.

Precipitation alone has a significant effect on materials and construction productivity [16,25], both during and after periods of rain. The lengths of the resulting delays are typically proportional to the amount of precipitation and the duration of the event [25],

with work stoppages resulting if precipitation rates become too high. Estimates of reduction in productivity resulting from precipitation events range between 40% for light precipitation [26] to upwards of 60% [27]. Productivity is also affected by both low and high air temperatures [25]. High temperatures may lead to heat stress or the dehydration of workers [25], and work stoppages are recommended when temperatures decrease below $-25\ °C$ to ensure worker safety [26]—particularly when low temperatures are combined with high wind speeds (measured as wind chill) [16]. Wind alone can result in work stoppages, decreased productivity, and negative impacts on materials. High wind conditions may cause the surface of fresh concrete to dehydrate and crack, and can increase the risk of accidents when performing high-altitude work [25]. Lifting activities are also affected by wind, with the impact depending on a combination of factors, such as wind speed, height of the lifting operation, and the type of object to be lifted [25].

Wind farm projects require locations with relatively high and consistent wind speeds to maximize electricity generation during the operation phase of the project. From a contractor's perspective, however, high wind speeds represent a significant risk in terms of safety, time, and cost [2,3]. Installation of the turbine and tower sections of a structure is completed using heavy cranes. In addition to effects on worker and equipment productivity, safety regulations mandated by many regulatory bodies require lifting activities to be halted at certain wind speed thresholds to avoid crane overturning [3]. Reductions in productivity of turbine installation due to wind speed have been detailed by Guo, Chen, and Chiu [2].

The impact of weather on construction projects is highly variable, depending on numerous factors, including the type of construction, location, and season [28], and affecting the individual tasks of a construction project differently. Low temperature, rain, and other weather conditions may hamper productivity of certain workers and machinery, while others may be unaffected. Regulations mandating threshold values for weather conditions at which work must stop varies between countries and even from city to city [29]. Given the uncertainty in weather and the variability associated with its impact, appropriately considering weather as a risk factor requires the use of sophisticated risk analysis and decision-support tools before and during construction execution [13]. To achieve the planned schedule, advanced planning is required to prepare mitigation strategies that can effectively reduce the negative impact of weather on project cost and time [25,29,30].

Monitoring and control tools are commonly used to evaluate project performance during the execution of construction projects. According to the PMBOK® Guide (2008) [31], project monitoring and control consists of a set of processes designed to track, review, and regulate the progress and performance of the project, identify areas in which changes to the plan are required, and initiate corresponding changes [31]. Continuous monitoring and control provide the project team with insights into the health of the project to identify any areas that may require additional attention [31], including project cost, schedule, quality, risk, scope, procurement, and communication management. Engaging in timely project monitoring and control can help minimize deviations from planned baselines to achieve original project objectives [32–35]. Lookahead scheduling represents one of the decision functions within production control systems [11]. This approach is often implemented in coordination with other monitoring and control tools.

Project monitoring and control tools were classified as either one-dimensional or multi-dimensional control systems [34]. One-dimensional project control tools are simple to implement; however, control is focused on one specific dimension (i.e., cost) rather than on the entire set of project objectives [34]. Multi-dimensional project control tools integrate several dimensions within one control system, such as the earned value (EV) approach, which was designed to assess cost and time simultaneously [34]. A detailed review of the tools proposed by different bodies of knowledge for project monitoring and control was compiled by Montes-Guerra [33]. Recent advances in this field include the automation of project monitoring and control to reduce the time required to obtain as-built data through the introduction of digital technologies, such as sensors, GPS, and RFID [35–37].

### 2.2. Weather-Related Project Delays in Construction

"Clear and specific" weather-related clauses are a common component of wind farm construction contracts [24]. Designed to allocate responsibilities and reduce claims, these clauses clarify compensation for weather delays caused by productivity loss or work stoppages [29]. Construction contracts generally differentiate between weather delays that can be anticipated and those that cannot [22]. Delays resulting from severe weather that is anticipated is usually considered non-excusable, where only delays caused by abnormal and unforeseeable weather events are granted a time extension [18,29]. Many wind farm construction projects are awarded by the owner on a calendar-day basis [2]. To determine the expected project duration for bidding, contractors typically use approximation or quantitative methods to calculate the number of days expected to be impacted by severe weather for bidding (and other pre-construction) purposes. Often, the expected number of non-working days due to weather are specifically defined in the contract [16], with severe penalties imposed on the contractor for projects that are not completed by the contract-specified end date.

Typically, approximation methods are used to determine the number of severe weather-associated days for bidding and pre-construction planning. Approximation methods involve the review of historical weather data to calculate the average number of days associated with severe weather conditions each month and using the remaining working days to develop the project schedule [29]. In addition to approximation methods, numerous quantitative methods have been developed. Quantitative methods use historical weather data to evaluate the impact of weather on project schedules, either directly or through a weather generator, as shown in Table 1. Weather generators are numerical models that reproduce synthetic weather data as a daily time-series of weather variables with the same statistical properties as historical weather data [29]. Both parametric and non-parametric approaches are applied. While most models use simulation, others have adopted a mathematical approach or a combination of fuzzy logic with the critical path method (CPM). Three weather parameters have received the most attention: temperature, precipitation, and wind speed. Previous studies have either modeled the effect of one weather parameter or a combination thereof.

**Table 1.** Summary of quantitative weather models.

| Reference | Modeling Theory [1] | Project Type | Weather Modeling | Weather Parameters [2] | | |
|---|---|---|---|---|---|---|
| | | | | Temp. | Wind | Prec. |
| [17] | Fuzzy + CPM | Highway | Historical | - | - | ✔ |
| [13] | Fuzzy + CPM | Highway | Historical | - | - | ✔ |
| [2] | Fuzzy + CPM | Onshore Wind | Historical | - | ✔ | - |
| [38] | Mathematical | Onshore Wind | Historical | - | ✔ | ✔ |
| [14] | Mathematical | Highway | Historical | - | - | ✔ |
| [16] | Mathematical | Highway | Generator | - | - | ✔ |
| [24] | Mathematical | Buildings | Historical | ✔ | ✔ | ✔ |
| [39] | Mathematical | Buildings | Historical | ✔ | ✔ | ✔ |
| [40] | Mathematical | Bridge | Historical | ✔ | ✔ | ✔ |
| [3] | DES | Onshore Wind | Generator | - | ✔ | - |
| [23] | DES | Tunneling | Generator | ✔ | ✔ | ✔ |
| [41] | DES | Dams | Generator | - | - | ✔ |
| [29] | DES | Tall Buildings | Generator | ✔ | ✔ | ✔ |
| [25] | DES | In Situ Wall | Historical | ✔ | ✔ | ✔ |
| [15] | DES + Continuous | Pipeline | Generator | ✔ | ✔ | ✔ |
| [42] | M.-Crit. + Reg. | Residential | Historical | ✔ | ✔ | ✔ |

[1] Fuzzy + CPM: fuzzy logic and critical path method, DES: discrete-event simulation, M. Crit. + Reg.: multi-criteria and regression; [2] Temp.: temperature, Prec.: precipitation.

### 2.3. Assessing the Impact of Weather on Wind Farm Construction

Of the quantitative weather models previously developed, only three were designed for onshore wind farm construction: Guo et al. [2] proposed a fuzzy-based approach for assessing the impact of wind speed uncertainty on wind turbine installation; Atef et al. [3] introduced a discrete-event simulation approach of wind turbine assembly activities coupled with a weather generator; and Zhou et al. [38] proposed a mathematical optimization approach to optimize the schedule under resources constraints in consideration of wind speed and precipitation. Notably, only the effect of wind speed and precipitation on turbine assembly was considered by these three studies, and all studies were designed to generate a holistic, master scheduling of the project at the early planning stage. Although useful for early scheduling during the pre-construction phase, these approaches are not suitable for developing short-term lookahead schedules used during the execution phase.

Currently, lookahead scheduling in wind farm construction relies on conventional scheduling techniques, such as bar charts and the CPM. However, these techniques are not able to precisely capture the impact of weather uncertainties or to model productivity-influencing factors when developing short-term project schedules [2]. In practice, the impact of wind uncertainty is usually estimated by a rule-of-thumb approach and subjective judgements based on practitioners' past experiences [2]. This approach can result in inappropriate adjustments, which may lead to deviations from the planned schedule [2]. Tools capable of reliably quantifying—in a detailed manner—the schedule delays associated with adverse weather are expected to result in improved management of weather risks, more realistic scheduling, enhanced utilization of construction resources, and safer work environments [3]. Reliable quantitative tools for short-term lookahead scheduling in wind farm construction, however, have yet to be reported in construction engineering and management literature.

### 2.4. Research Gaps

Barriers limiting the application of existing quantitative methods to assess the impact of weather in onshore wind farm construction for short-term lookahead scheduling include:

1. Methods for assessing the impact of weather in onshore wind farm construction [2,3] only addressed the impact of wind speed on turbine installation and did not consider the influence of other weather parameters, such as precipitation and air temperature, on project schedules.
2. Existing methods [2,3] were limited to turbine installation, and could not consider the impact of weather on other construction activities. To examine the impact of weather on the project schedule, all project activities and their criticality should be modeled and considered. This is particularly important when considering that certain non-critical activities may become critical as a result of weather delays. Conversely, certain weather-sensitive activities may not fall on the critical path, with weather-related delays in these instances not affecting project duration. For example, a weather-related delay in pouring the concrete foundation will delay all subsequent construction activities, resulting in a considerable impact on overall project duration. In contrast, a weather-related delay in the non-critical activity of substation drainage installation will have a minimal impact on the overall project duration
3. Short-term weather forecasts are typically more accurate and reliable than historical weather data [43]. Existing methods, presented in Table 1, use historical weather data as an input—either directly or through a weather generator—which often results in daily weather predictions that are not matched to actual weather conditions during the short-term lookahead period.
4. Existing methods, presented in Table 1, are unable to incorporate as-built data into the quantitative scheduling system as the project progresses, thereby limiting the accuracy and representativeness of the output schedules during the construction phase.

*2.5. Simulation as a Proposed Approach*

Construction simulation allows for the development of and experimentation with computer-based representations of construction projects at a detailed level to understand their underlying behavior and investigate the effects of external factors [44]. The ability of discrete-event simulation (DES) to incorporate the variability associated with external factors, such as weather, to determine the impact of uncertainty on system outcomes is well-established. As presented in Section 2.2 (Table 1), several studies have successfully applied DES to investigate the effects of adverse weather on construction activities [3,15,23,25,29,41], and DES was shown to be a reasonable tool for scheduling construction activities of onshore wind projects [45]. However, a simulation model capable of considering the impact of weather on the activities on onshore wind farm construction has not been developed.

## 3. Proposed Framework

This research is proposing a hybrid DES-continuous simulation-based framework to integrate as-built information with short-term precipitation, air temperature, and wind speed forecast data to improve lookahead scheduling at the project-level in onshore wind farm construction. The proposed framework centers around a newly-developed and generic hybrid simulation model (i.e., discrete-event and continuous) of both weather-sensitive and non-sensitive activities in onshore wind farm construction. Advances to the existing state-of-the-art include the ability of the model to simultaneously: (1) consider the influence of additional weather parameters, such as precipitation and air temperature, on project schedules, (2) model all critical and non-critical construction activities, (3) incorporate short-term weather forecast information, and (4) integrate as-built data to enhance short-term lookahead scheduling in onshore wind farm construction.

The proposed framework consists of three components: (1) data collection and preparation, (2) simulation, and (3) framework outputs, as shown in Figure 1. First, the method examines short-term weather forecast data and determines upcoming weather conditions over the lookahead period (e.g., 14 days). The productivity of any uncompleted activities during the lookahead periods are multiplied by a pre-established productivity factor to determine a new activity duration given the expected weather.

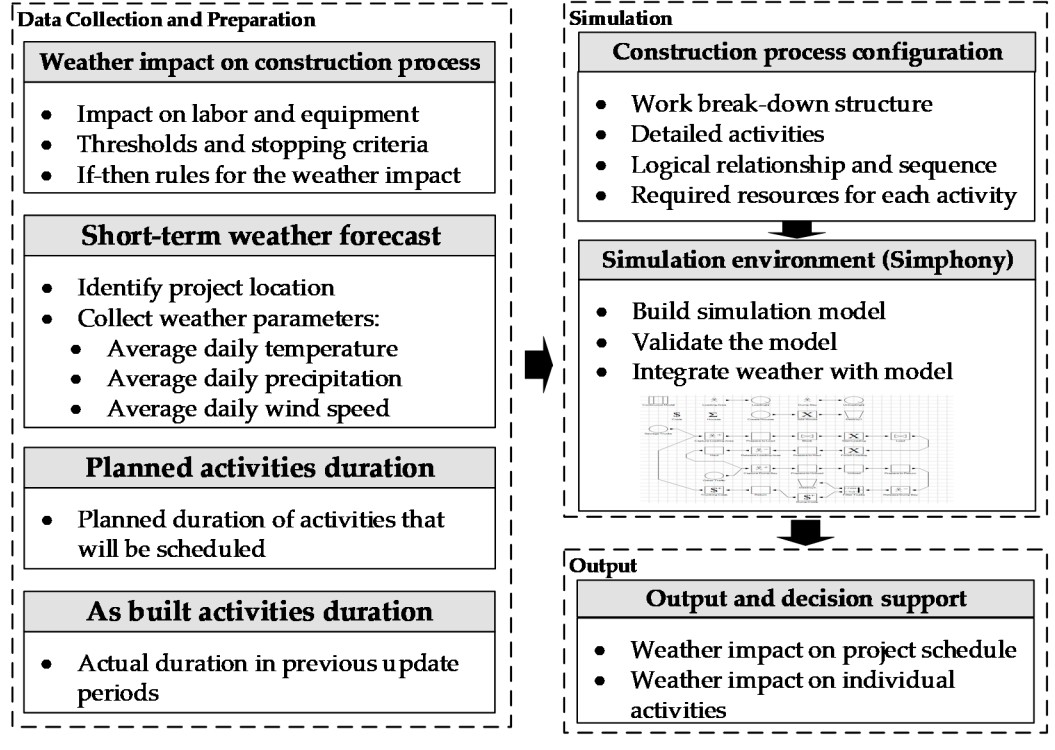

**Figure 1.** Proposed framework.

Using the new durations for the activities within the specified lookahead period together with (1) the actual durations of completed activities and (2) the planned durations of activities in the post-lookahead period, the method generates an updated project schedule each time new as-built information is entered (Figure 2). This process is repeated each lookahead update period (i.e., i, i + 1, . . . .) until the project is completed.

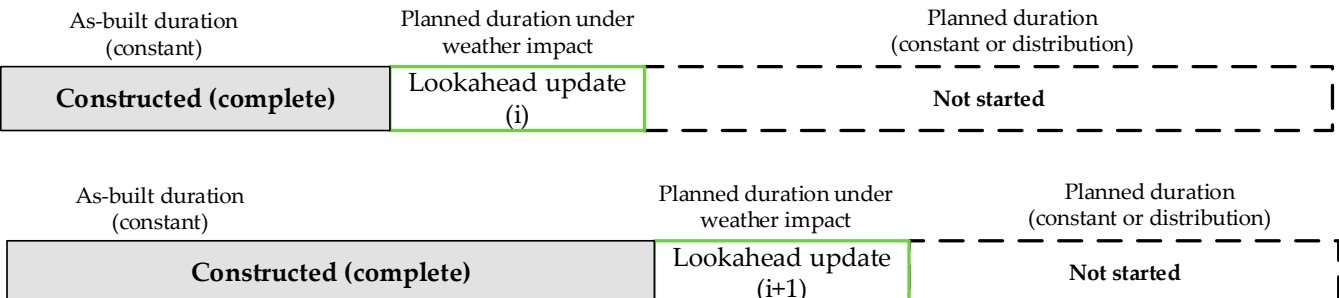

**Figure 2.** As-built versus planned durations for two lookahead update periods.

*3.1. Data Collection and Preparation*

The inputs required to apply the proposed approach include: (1) the weather impact on productivity of construction activities, (2) the short-term weather forecast for the lookahead period, (3) the planned duration of activities not yet completed, and (4) the as-built (i.e., actual) duration of completed activities.

3.1.1. Weather Impact on Productivity

This input associates weather parameter values to construction productivity using if-then rules encompassing three considerations [15]: (1) weather parameters that influence the activity, (2) weather conditions that would cause each activity to stop (i.e., stopping conditions), and (3) the relationship between activity productivity and weather conditions.

First, activities are listed, and whether or not each activity is sensitive to weather is determined. Then, the specific weather parameters that are capable of affecting productivity are identified for each weather-sensitive activity. For example, turbine assembly is sensitive to wind speed because of the crane lifts associated with this activity, whereas labor-dependent activities are generally influenced by both temperature and wind speed.

Then, thresholds for each weather parameter for each activity are determined. Thresholds are divided into weather-related work stoppage thresholds, which define the weather values beyond which the construction activities cannot proceed [15,18,21,25,29], and weather-related productivity loss thresholds, which define the weather values at which construction work can continue, but at a lower productivity [18,21]. Threshold values are affected by a number of factors, such as trades and natural and social factors, including location of the project and cultural mores (i.e., unofficial norms) recommending weather conditions at which trades can work [18]. Official threshold values, when available, can be obtained from a variety of sources, including work safety regulations, organization-specific practices, historical data, subject matter experts, and/or construction literature. In some cases, work stoppage thresholds can be written into construction contracts to reduce disputes that may arise due to weather-related work stoppages [18,29].

Finally, for activities with productivity loss thresholds, a mathematical relationship between weather parameter values and productivity is established for each set of weather parameter values. Two methods for deriving a mathematical relationship were reported in literature. The first method calculates the impact on activity duration directly (i.e., percentage of duration is added to the original duration [3]), while the second method calculates the impact of weather on activity productivity through a productivity factor [15]. Productivity factor values can be calculated by inputting historical project data and/or subjective information into the productivity factor equation proposed by [15]. Notably,

in the absence of applicable historical information, productivity factor values can also be obtained from construction literature. In adverse conditions, productivity factors are less than a value of 1; while in favorable conditions, productivity factors are higher than a value of 1.

### 3.1.2. Short-Term Weather Forecasts

Short-term weather forecasts are available from a variety of publicly-available and commercial providers, which typically provide daily weather forecasts for the upcoming 7, 10, and/or 14 days. Weather conditions encompass a number of parameters, including sky temperature, atmospheric pressure, cloudiness, humidity, air temperature, wind speed, and precipitation. An extensive review of weather-related construction delays, compiled by Schuldt et al., "identified extreme temperatures, precipitation, and high winds as the most impactful weather conditions on construction" [21]. These findings were confirmed by the practitioners in Section 4.1.1, who asserted that, based on their experience, wind speed, air temperature, and precipitation were the most critical factors affecting productivity of onshore wind farm construction projects.

### 3.1.3. Activity Durations: Planned and As-Built

The planned durations for activities that have not yet been completed are determined using historical project data or subject matter expert opinion. At the initial stage of construction, all activities will be input with planned durations. Methods for determining activity durations are extensively discussed in construction literature. Readers are referred to [46] for a detailed review of activity duration planning. Planned durations can be input as constant values or as probability distributions. As activities are completed, planned durations will be replaced with as-built (i.e., actual) durations within the simulation model. This applies to both weather-sensitive and non-sensitive activities. As-built durations can only be input as constant values, given that as-built durations are known.

### *3.2. Simulation*

Once the required data are collected, they are input into the proposed simulation model for onshore wind farm construction. Here, DES was used to model non-sensitive weather activities, and—due to dynamic changes in weather conditions—continuous simulation was used to model weather-sensitive activities. The model is capable of considering both regular variability (through the input of planned durations as probability distributions) and the impact of weather on productivity (through the application of the productivity if-then rules) to predict durations of both the individual activities and of the overall project. Model development and application are detailed as follows.

### 3.2.1. Model Development

A simulation model can be developed once the underlying system behavior is defined and understood. The construction process of onshore wind projects and simulation logic are detailed as follows.

- Construction Process Configuration;

To develop the model, common components of onshore wind farm construction were identified and abstracted. Previous studies were reviewed [45,47,48], and a typical onshore wind farm project was found to be comprised of six major work packages: site preparation, the foundation, turbine assembly, the collection system, mechanical completion, and commissioning. Each of these work packages was further partitioned into more detailed work-packages, as shown in Figure 3. As the proposed method requires weather impacts on activity duration to be determined, the work packages were further partitioned at the activity level following a detailed review of: (1) previous onshore wind farm construction projects available in literature [2,3,45,47] and (2) 10 real onshore wind farm projects, as detailed in Supplementary Materials (Table S1).

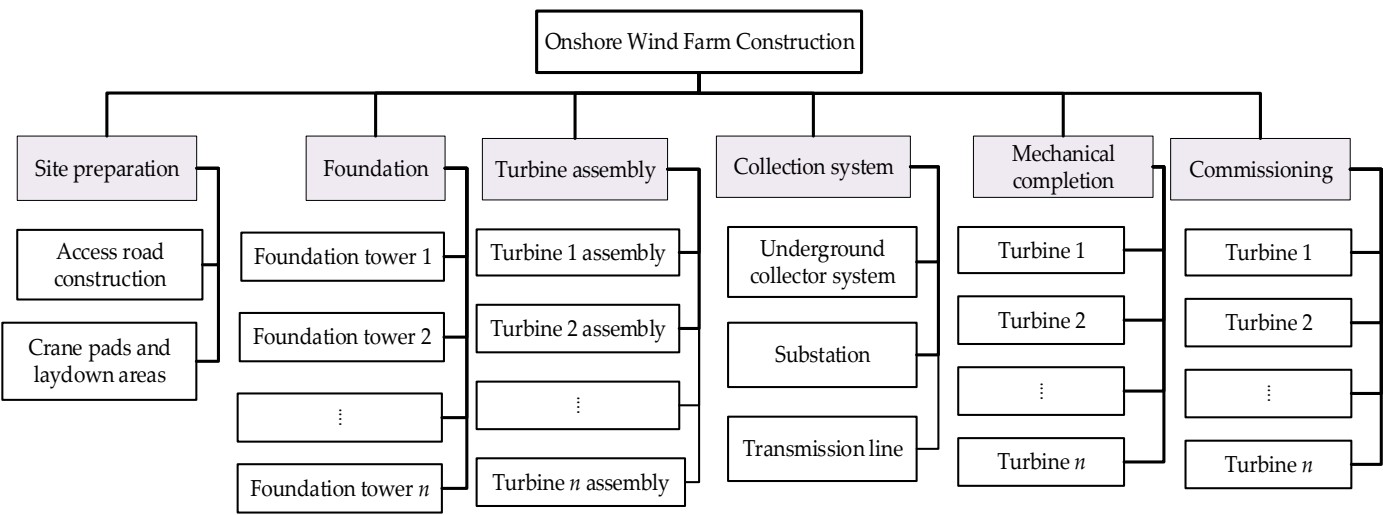

**Figure 3.** Work breakdown structure of an onshore wind project.

The detailed activities within each work package, and the logical relationships between them, are illustrated in Figure 4. The generic activity-level WBS (Figures 3 and 4) was reviewed by subject matters experts, who confirmed that the WBS was accurate and representative of a typical onshore wind farm construction project.

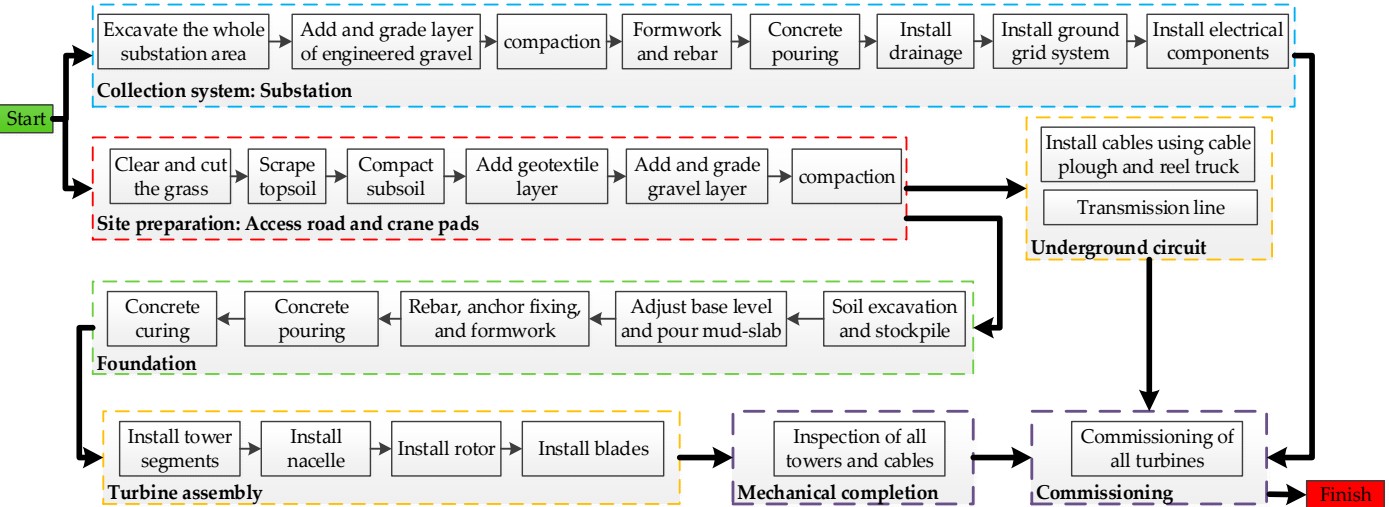

**Figure 4.** Detailed activities and sequence of work for an onshore wind project.

- Simulation Logic;

Once the generic construction process was established, a combined discrete-event and continuous simulation modeling approach was used to develop a generic simulation model. The simulation logic underlying the hybrid simulation model is detailed in Figure 5. In DES, entities are objects that have attributes, experience events, consume resources, and enter queues over time. An entity can be dynamic by moving through the system or remain static to serve other entities [49]. As an entity moves through the model, events are scheduled, thereby representing the progress of the system.

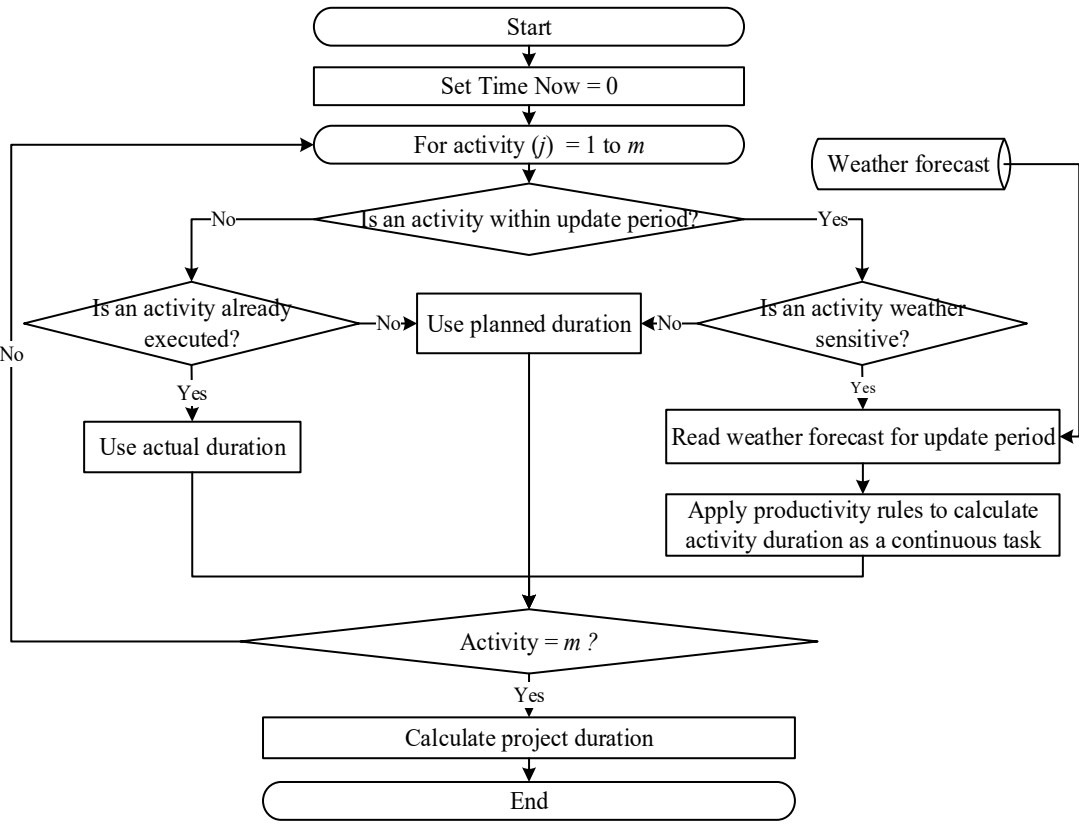

**Figure 5.** Simulation logic for activities and project duration calculation for an update period.

In the current study, one entity is created at the beginning of the simulation. Depending on the activity sequence, further entities are created as needed. Following the creation of an entity, the simulation model retrieves the forecasted weather parameter values for each day of the update period (i.e., 14 days). When an entity arrives at a weather-sensitive activity, one of three statuses is selected based on the "time now" value:

(1) If the "time now" value is greater than the lower boundary of the update period, but less than the upper boundary of update period, the weather-sensitive activity is included in the lookahead period. Continuous simulation, as was recommended in the literature [11,21], is then applied to model productivity in consideration of the weather impact.

(2) If the "time now" value is greater than the upper boundary of update period, the weather-sensitive activity does not fall within the lookahead period, thus the planned activity duration is used.

(3) If the "time now" value is less than the lower boundary of update period, the weather-sensitive activity was completed in the previous lookahead period, thus the actual duration is used.

A non-sensitive weather activity has only two statuses: completed or to be executed. For completed activities, actual durations are used. Planned durations are used for incomplete or to be executed activities.

The simulation model provides successive progress updates at one-day intervals. The partial completion of a weather sensitive activity is permitted at the end of an update cycle. Activities with a duration greater than the lookahead period (i.e., 14 days) are divided into segments with durations equal to or less than the lookahead period. The progress for discrete activities is assessed based on the number of discrete units completed and the time required for their execution. This process continues until the project is fully simulated. Based on the simulation results, a total project duration is calculated. The simulation

model is run for a pre-determined number of iterations, and the results of each iteration are combined to provide final results.

### 3.3. Framework Outputs

The first output of the model is the expected duration of the individual weather-sensitive activities in consideration of the short-term weather forecast. This is presented as a plot that visualizes the progress of activities and total accumulated duration. The second output of the model is the expected duration of the entire project. This is visualized as a histogram, illustrating the range and distribution of project durations obtained from each run of the simulation model. Various statistics can also be obtained from the simulation results, including minimum, average, and maximum durations, as well as the standard deviation. Lastly, the model allows tracking and extraction of finish times for individual activities, which can be visualized as a histogram.

These outputs can provide effective, proactive decision-support to practitioners, helping activities (particularly those on the critical path) remain on schedule and reducing the likelihood of project delays. A discussion of the practical implications of the framework outputs are detailed in Section 5.1.

## 4. Case Study

A real wind farm project was used to demonstrate the functionality and applicability of the proposed framework. The onshore project consisted of eight 5.0 MW wind turbine generators for a total project output of 40 MW. The focus of the case study was on construction activities that were of interest to our industrial partner. Other activities, such as electrical tasks, were not included in this study. Data from two points (i.e., update periods) during construction execution were collected and used to evaluate the ability of the framework to incorporate as-built data.

### 4.1. Data Collection and Preparation

4.1.1. Weather Impact on Productivity

First, rules that describe the impact of weather parameters on productivity were developed. Influencing weather parameters and threshold conditions for each activity were collected from construction literature or were provided by the contractor. The input data were reviewed by experts in the field, who confirmed that the thresholds were appropriate for the jobsite. The input data, together with their sources, are summarized in Tables 2 and 3. The qualifications of the subject matter experts are listed in Table 4. Importantly, the experts indicated that work must also be suspended for approximately five working days following a heavy snowfall to ensure that all materials, including blades and other components, are cleared of snow.

**Table 2.** Weather-sensitive activities and influencing weather parameters.

| Activity | Weather Parameters | | |
| --- | --- | --- | --- |
| | Temperature | Wind | Precipitation |
| Excavation | ✔ | | ✔ |
| Compaction | ✔ | | ✔ |
| Formwork and rebar | ✔ | ✔ | ✔ |
| Concrete pouring | ✔ | ✔ | ✔ |
| Install tower segments | ✔ | ✔ | |
| Install nacelle | ✔ | ✔ | |
| Install rotor | ✔ | ✔ | |
| Mechanical completion | ✔ | ✔ | |
| Commissioning | ✔ | ✔ | |



**Table 3.** Stopping thresholds for weather-sensitive activities.

| Activity | Weather Parameters | | | Reference |
|---|---|---|---|---|
| | Temperature (°C) | Wind (m/s [1]) | Precipitation (mm/h) | |
| Excavation | <−25 | - | >5 | [16] |
| Compaction | <−25 | - | >5 | [16] |
| Formwork and rebar | <−25 | >15 | >5 | [16,25] |
| Concrete pouring | <0 | >11.5 | 2.5 * | [16,25] |
| Install tower segments | <−25 | >14 | - | [2,3] |
| Install nacelle | <−25 | >14 | - | [2,3] |
| Install rotor | <−25 | >14 | - | [2,3] |
| Mechanical completion | <−25 | >11 | - | Expert |
| Commissioning | <−25 | >11 | - | Expert |

[1] 1 m/s = 3.6 km/h. * During precipitation event.

**Table 4.** Expert qualifications.

| No. | Years of Experience in Industry | Education Level |
|---|---|---|
| 1 | 8 | Doctorate |
| 2 | 15 | Master |
| 3 | 7 | Bachelor |

As historical project data were not available, relationships between productivity and weather parameters were obtained from those proposed in construction literature. Reported impacts of air temperature (Figure S1) [26], precipitation (Figure S2) [25], and wind speed (Figure S3) [2] on productivity from the literature were examined, previous projects were reviewed, and a list of rules was prepared based on these findings. A total of 150 rules were defined for this case study. As mentioned previously, the list of rules can differ between organizations and from project-to-project. As such, the list of rules should be reviewed and modified, if required, for each new project. A sample of the developed rules is summarized in Table 5. The complete list of rules was validated by the subject matter experts.

**Table 5.** Sample of developed rules.

| Weather Parameters | | | Productivity Factor |
|---|---|---|---|
| Temperature (°C) | Wind (m/s) | Precipitation (mm/h) | |
| T < −25 | P = 0 | W = 0 | 0 |
| −24.9 < T < −15 | P = 0 | W = 0 | 0.55 |
| −24.9 < T < −15 | 0 < P < 0.5 | W < 7 | 0.4 |
| −24.9 < T < −15 | 0.5 < P < 1 | W > 7 | 0 |
| −24.9 < T < −15 | 1 < P <4 | W >7 | 0 |
| −14.9 < T < −5 | P = 0 | W = 0 | 0.75 |
| −14.9 < T < −5 | 0 < P < 0.5 | 0 < W < 7 | 0.65 |
| −14.9 < T < −5 | 0.5 < P < 1 | 7 < W < 10 | 0.6 |
| −14.9 < T < −5 | 1 < P < 4 | 7 < W < 10 | 0.5 |

4.1.2. Short-Term Weather Forecasts

In this study, a 14-day weather forecast was selected, as it matched the lookahead update period used by the contractor, which was two weeks. Due to its easy-to-use interface, Dark Sky API [50] was used to collect weather data, including temperature, wind, and precipitation levels. Since weather forecast data were provided hourly, hourly values during the period of construction operations (8:00 AM to 5:00 PM) were averaged to obtain daily forecast values. Data for the two update periods (i.e., 28 days) was collected, as

detailed in Table S1. The project was located in Alberta, Canada; weather information for the project specific location was extracted.

### 4.1.3. Activity Durations

The third input is the planned durations and logical relationships of the activities under each work package, as shown in Table 6. Activity durations specific to this case study were provided by the contractor. Widely-used and commonly-recommended [47,51–53], triangular distributions were used to stochastically represent regular (i.e., non-weather) variability in construction activity durations. Notably, activities related to site preparation, foundation, turbine assembly, mechanical completion, and commissioning are repeated for each of eight turbines.

**Table 6.** Project activity details.

| Work Package [1] | Activity | ID | Duration (Days) [2] | Pred. ID/ Rel. (Lag) [3] | Required Resources [4] |
|---|---|---|---|---|---|
| Site Preparation | Scrape topsoil | A1 | Tri (5, 7, 6) | - | Bulldozer |
| | Compact subsoil | A2 | Tri (5, 7, 6) | A1/F.S | Compactor |
| | Add geotextile layer | A3 | Tri (2, 4, 3) | A2/F.S | Geotextile Crew |
| | Add and grade gravel layer | A4 | Tri (5, 7, 6) | A3/F.S | Grader |
| | Compaction | A5 | Tri (5, 7, 6) | A4/F.S | Compactor |
| Collection System: Substation | Excavate substation area | A6 | Tri (7, 12, 10) | A0/F.S | Excavator |
| | Add and grade gravel layer | A7 | Tri (3, 5, 4) | A6/F.S | Grader |
| | Compaction | A8 | Tri (2, 4, 3) | A7/F.S | Compactor |
| | Formwork and rebar | A9 | Tri (7, 12, 10) | A8/F.S | Crew |
| | Concrete pouring | A10 | Tri (1, 3, 2) | A9/F.S | Pouring Crew |
| | Install drainage | A11 | Tri (5, 12, 7) | A10/F.S | Crew, Excavator |
| Foundation Construct | Soil excavation | A12 | Tri (2, 3, 2.5) | A5/F.S | Excavator |
| | Adjust base level, pour slab | A13 | Tri (1, 2, 1.5) | A12/F.S | Pouring Crew |
| | Rebar, anchor, formwork | A14 | Tri (2, 4, 3) | A13/F.S | Crew |
| | Concrete pouring | A15 | Tri (1, 2, 1.5) | A14/F.S | Pouring Crew |
| | Concrete curing | A16 | 21 | A15/F.S | - |
| Circuit | Install cables | A17 | Tri (100, 110, 105) | A5/F.S | Cable Plough, Crew |
| Turbine | Install tower segments | A18 | Tri (2, 3, 2.5) | A16/F.S | Crane, Assy. Crew |
| | Install nacelle | A19 | Tri (0.5, 1, 1) | A18/F.S | Crane, Assy. Crew |
| | Install rotor and blades | A20 | Tri (2, 3, 2.5) | A19/F.S | Crane, Assy. Crew |
| Mechanical | Inspection of one tower | A21 | Tri (3, 7, 5) | A22/F.S | Crane, Insp. Crew |
| Commis. | Commissioning one turbine | A22 | Tri (5, 9, 7) | A11/F.S A17/F.S A21/F.S | Crew |

[1] Commis.: commissioning; [2] Tri: triangular distribution; [3] Pred.: predecessor activity, Rel.: relationship; [4] Assy.: assembly, Insp.: inspection.

### 4.2. Simulation

In this study, an in-house developed simulation engine, *Simphony.NET* 4.6 [54,55], was used as the simulation environment to model the onshore wind project activities and associated weather impact. The model was built using the approach detailed in Section 3.2. The unit of time was set to days, which included an 8 h workday and no night shifts. The developed rules were coded as if-then rules and stored using global variables in *Simphony.NET*, such that they were readable by all activities in the simulation model. Whether or not a weather-sensitive activity occurred during the update period was determined using a conditional branch in *Simphony.NET*, as shown in Figure 6. A snapshot of the entire model is provided in Figure S4.

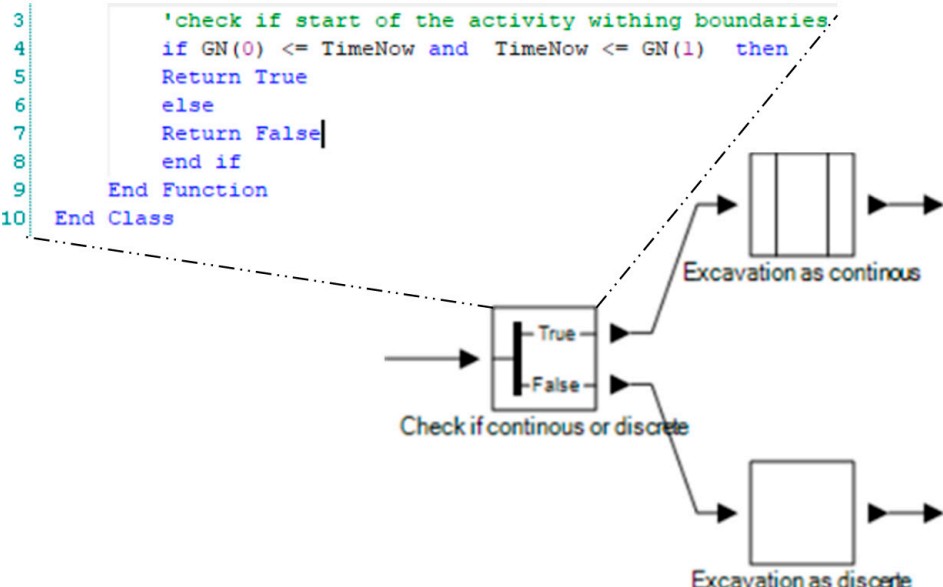

**Figure 6.** Modeling of weather-sensitive activity.

If a weather-sensitive activity did not occur during the update period (i.e., already completed or scheduled to begin in subsequent periods), the actual or planned duration was used, as described in Section 3.2. Weather data were input to a database that was read by the model every simulated day. For weather-sensitive activities occurring during the update period, the appropriate rule was identified, and the simulated activity duration was multiplied by the corresponding productivity factor.

The simulation was then run for 1000 iterations, as recommended in [56], to achieve the desired level of confidence. Notably, this was well in excess of the 120 iterations recommended for a simulation to reach maturity [57].

### 4.2.1. Simulation Model Validation

The generic hybrid simulation model was validated using trace validation, event validity, and face validation approaches [58]. First, trace validation, which records the behavior of various entities in a trace window, was used to evaluate the logic of the simulation model. The sequence of the activities through which the entities flowed was the same as the planned activity sequence, indicating that the logic of the simulation model was consistent with the logic observed in practice.

Second, using event validity, the simulated project duration was compared with the planned duration calculated by the contractor using commercial scheduling software [59]. The simulated project duration (without considering weather impact) was an average of 246 days ($\sigma = 4$), which was similar to the original deterministic project duration of 240 days, demonstrating that the model was capable of generating results that were representative of real events.

Third, face validation of the model's logic was conducted. Three subject matter experts, whose qualifications are listed in Table 4, reviewed the simulation logic of the model. All three experts confirmed that the logic was sound. Based on the findings of the trace validation, event validity, and face validation tests, the model was applied to the case study.

### 4.3. Framework Outputs and Results

The framework was applied to each of the two lookahead update periods. During the first lookahead update period, activities of two work-packages—site preparation and collection system—were initiated in this lookahead period. Of the two work packages, only two weather-sensitive activities (i.e., compaction of subsoil in site preparation and excavation of substation in collection system) were initiated during the first lookahead period. The planned duration of compaction was triangular (5, 7, 6) (Table 6) and excavation was triangular (7, 12, 10) (Table 6). The planned durations were used for the remainder of the activities scheduled to be executed during the first update period.

Weather conditions during the first update period were favorable (Table S2), with air temperatures ranging between 14.3 °C to 23.4 °C, wind speeds remaining below 8.75 m/s, and precipitation rates below 0.5 mm/h for 13 out of the 14 days. As expected, the impact of weather on the productivity of the weather-sensitive activities was minimal, ranging between 0.9 and 1.0 for both compaction (Figure 7a) and excavation (Figure 8a). The average simulated duration for compaction was 5.5 days, while the average simulated duration of excavation was 10.5 days. The accumulated duration for one simulation run of both activities are illustrated in Figures 7b and 8b. The impact of weather resulted in a simulated finish time (considering the impact of weather on productivity) for compaction of 11.5 days ($\sigma = 1$; Figure 9b) and excavation of 11 days ($\sigma = 1$; Figure 10b), which is similar to the finish time obtained when weather impact was not considered (Figures 9a and 10a). The average simulated total project duration was determined to be 246 days ($\sigma = 4$; Figure 11) compared to the planned project duration of 240 days. Since weather had a negligible impact on project duration, practitioners determined that no corrective actions were needed for this lookahead period.

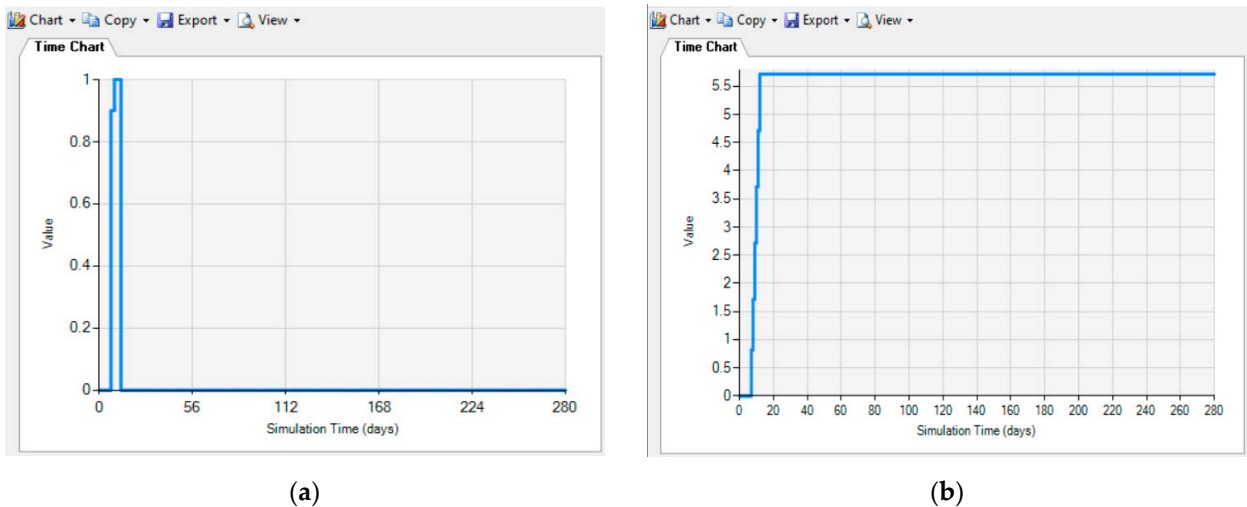

(a)            (b)

**Figure 7.** Weather impact on compaction activity: (**a**) progress as productivity factor, and (**b**) accumulated activity duration.

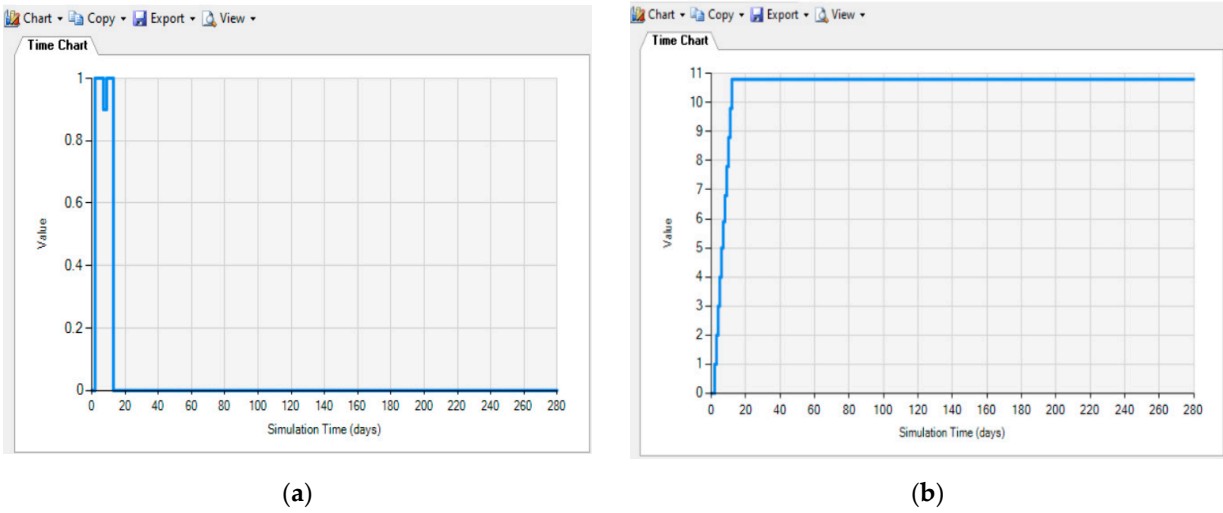

**Figure 8.** Weather impact on substation excavation activity: (**a**) progress as productivity factor, and (**b**) accumulated simulated activity duration.

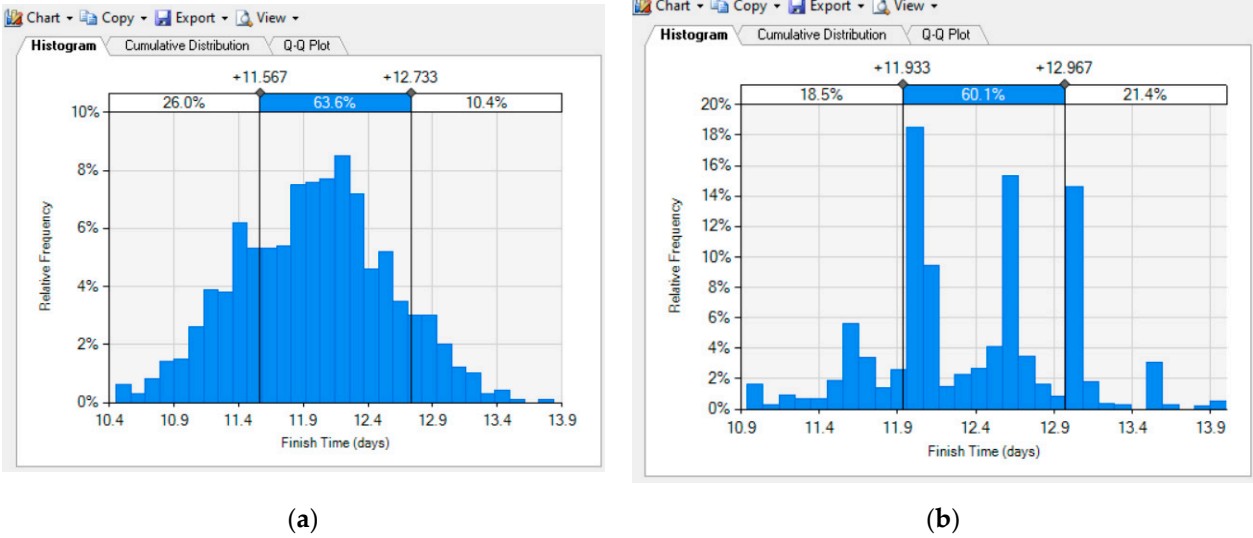

**Figure 9.** Finish time of compaction activity: (**a**) without weather impact, and (**b**) considering weather impact.

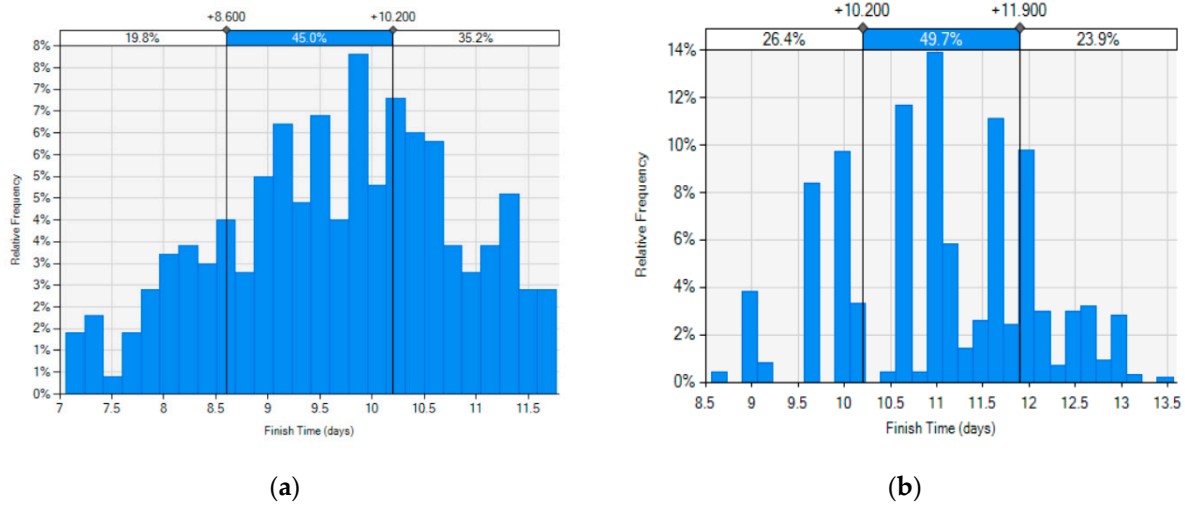

**Figure 10.** Finish time of excavation activity: (**a**) without weather impact, and (**b**) considering weather impact.

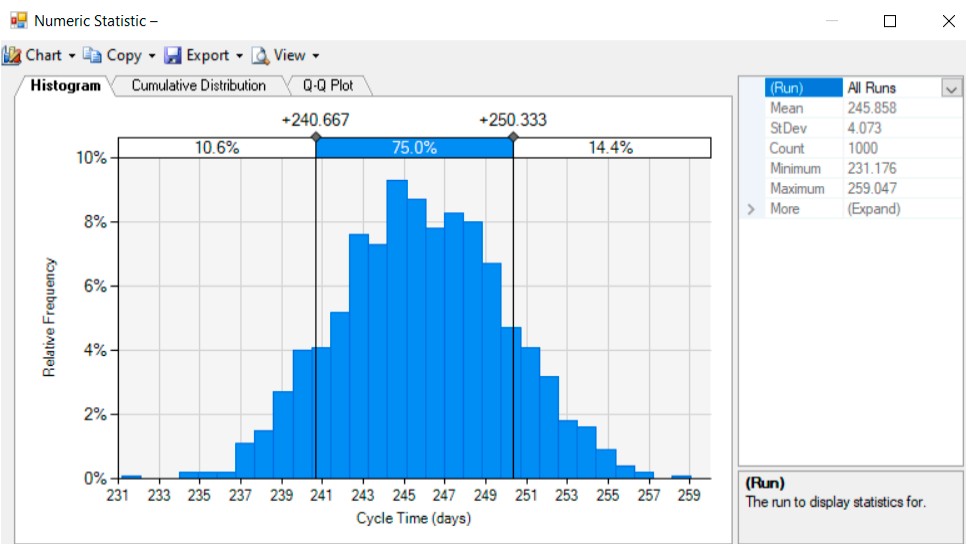

**Figure 11.** Project duration after considering weather impact for first lookahead update.

Similar to the first update period, weather conditions during the second update period were favorable (Table S2), ranging between 14.9 °C and 24.6 °C, with wind speeds remaining below 6.16 m/s, and no precipitation. Nine activities were initiated during the first update period, of which two were still ongoing. Progress information is detailed in Table 7.

**Table 7.** Progress of project activities at day 14.

| Activity | Weather-Sensitive? | Progress (%) | Actual Duration (Days) | Remaining |
|---|---|---|---|---|
| Scrape topsoil section 1 | No | 100 | 6 | 0 |
| Scrape topsoil section 2 | No | 100 | 5 | 0 |
| Compact subsoil of section 1 | Yes | 100 | 6 | 0 |
| Scrape topsoil section 3 | No | In Progress | 2 | Tri (3, 5, 4) |
| Compact subsoil of section 2 | Yes | In Progress | 3 | Tri (2, 4, 3) |
| Add geotextile layer of section 1 | No | 100 | 3 | 0 |
| Excavation of substation area | Yes | 100 | 11 | 0 |
| Gravel layer of substation | No | 100 | 4 | 0 |

The actual durations were used for completed activities. For in progress activities, a discrete task was used for the completed portion of the activity, and the remaining portion was modeled as a continuous task to allow for the consideration of weather effects. In addition to the in progress activities, seven weather-sensitive activities were initiated during the second update period. As expected, the favorable weather conditions had a minimal impact on productivity, resulting in a simulated project duration of 245 days ($\sigma = 4$; Figure 12). A high-level summary of the results for update periods 1 and 2 is provided in Table 8.

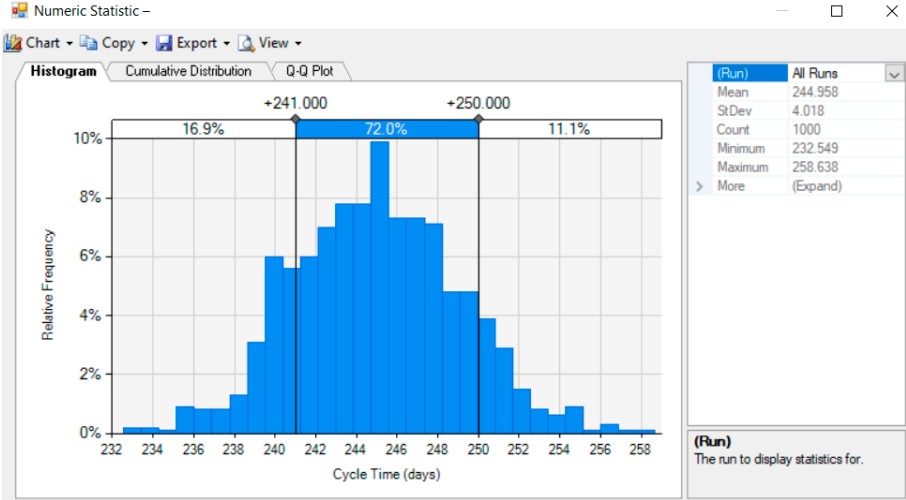

**Figure 12.** Project duration after considering weather impact for second lookahead update.

**Table 8.** High-level summary of results for update periods 1 and 2 of case study.

| Period | Impact on | | Project Duration | Corrective Action |
| | Productivity [1] | Total Project Duration | | |
|---|---|---|---|---|
| Baseline | - | - | 246 days ($\sigma$ = 4) | - |
| 1 | <10% | Minimal | 246 days ($\sigma$ = 4) | None required |
| 2 | <10% | Minimal | 245 days ($\sigma$ = 4) | None required |

[1] Weather-sensitive activities.

Framework Evaluation

In addition to validating the model's logic (detailed in Section 4.2.1), the framework was validated through two additional tests: a sensitivity analysis, and a face validation of the output results [58]. First, the sensitivity of the model to changes in weather conditions was examined. In contrast to favorable weather conditions observed in the case study, the sensitivity analysis input unfavorable weather data into the simulation model (Table 9). Here, air temperatures were below 0 °C, with average wind speeds ranging between 2 and 20 m/s, and precipitation present on 7 of the 14 days.

**Table 9.** New weather parameters for first 14 days.

| Days Since Start | Average Temperature (°C) | Average Precipitation (mm/h) | Average Wind Speed (m/s) |
|---|---|---|---|
| 1 | −10.0 | 1.00 | 2 |
| 2 | −3.5 | 2.00 | 4 |
| 3 | −7.7 | 3.50 | 2 |
| 4 | −8.0 | 1.50 | 3 |
| 5 | −9.4 | 4.00 | 5 |
| 6 | −9.4 | 2.00 | 6 |
| 7 | -8.9 | 0.00 | 7 |
| 8 | −12.1 | 0.00 | 8 |
| 9 | −16.8 | 5.00 | 9 |
| 10 | −1.1 | 0.00 | 14 |
| 11 | −1.9 | 0.00 | 15 |
| 12 | 0.0 | 0.00 | 20 |
| 13 | −1.9 | 0.00 | 8 |
| 14 | −1.8 | 0.00 | 6 |

The daily productivity factor of compaction ranged from 0.5 to 0.9, which, in turn, extended the duration of compaction from an average simulated planned duration of 6 days (Figure 13b) to a simulated weather-impacted duration of 11 days. This was reflected in the finish time of the activity, as shown in Figure 14. Altogether, the unfavorable weather conditions during the first 14 days alone caused an overall project delay of 5 days, resulting in a simulated project duration of 251 days (σ = 4) (Figure 15). Moreover, Figure 13a demonstrates that the sensitivity analysis results were consistent with expected outcomes, with unfavorable weather conditions resulting in longer activity durations and delayed project completion dates.

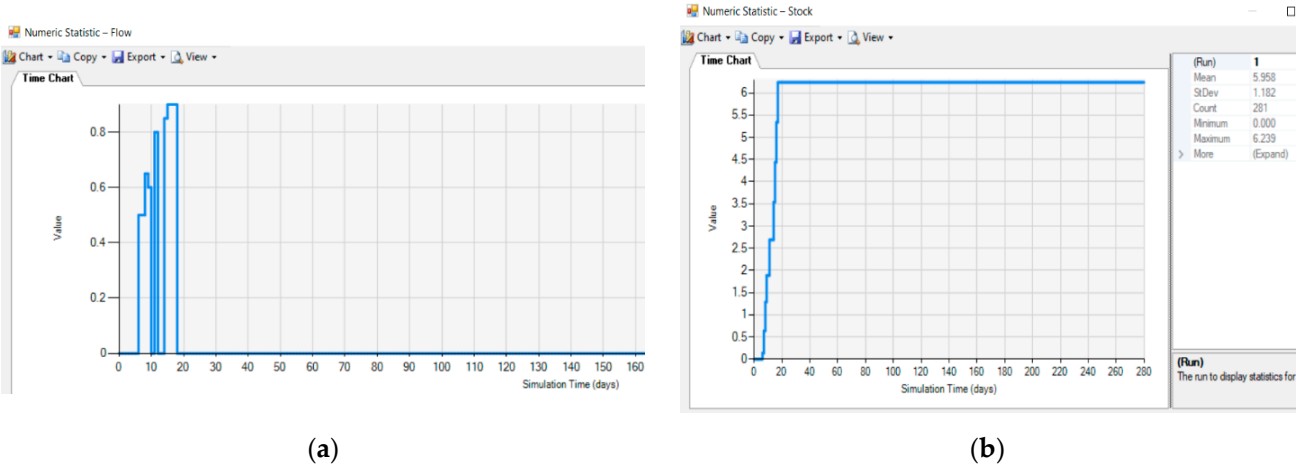

(**a**)                                   (**b**)

**Figure 13.** Weather impact on compaction activity: (**a**) progress as productivity factor, and (**b**) accumulated activity duration.

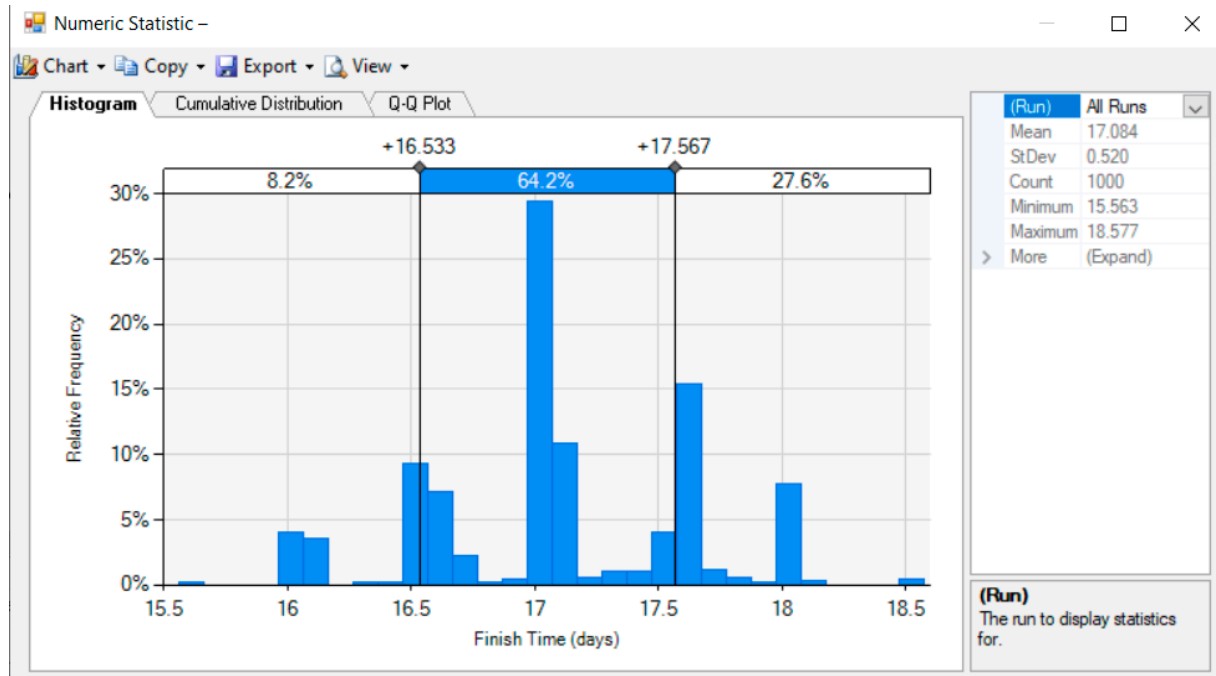

**Figure 14.** Finish time of compaction activity under new weather parameters.

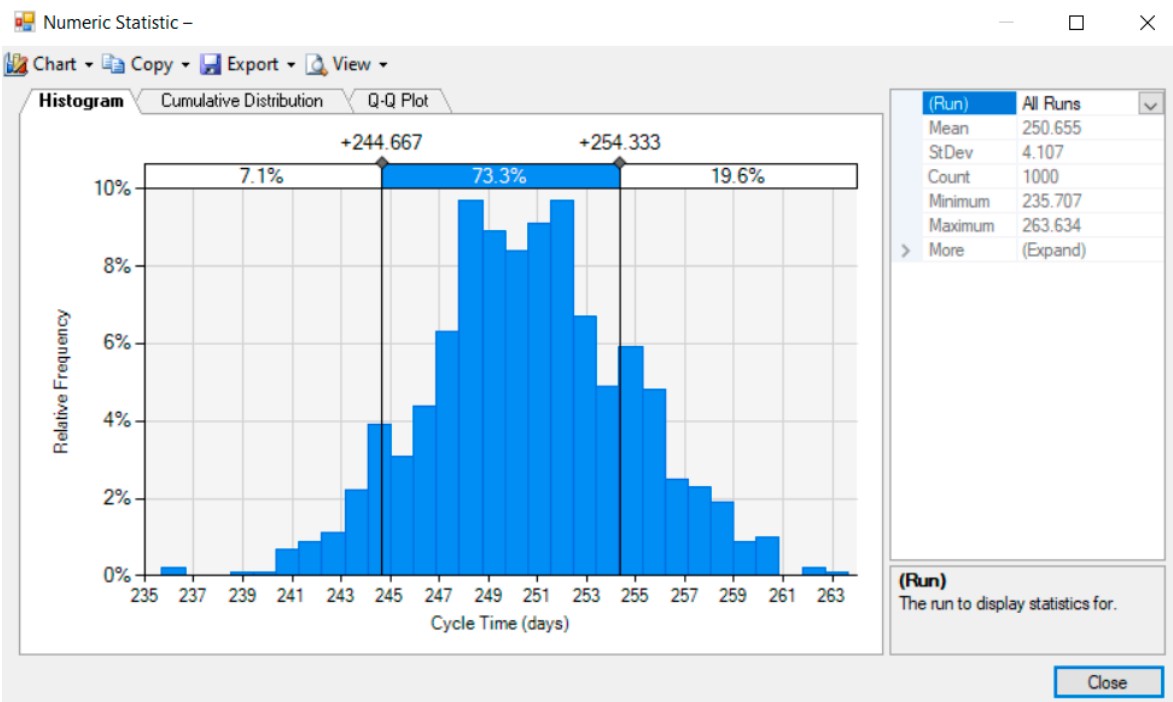

**Figure 15.** Project duration under impact of new weather parameters.

Table 10 presents a comparative summary of the first lookahead update period and the baseline schedule. Notably, poor weather conditions, causing a 50% reduction in the productivity of weather-sensitive activities, resulted in a 5-day delay in project completion time. Corrective actions would be required to mitigate the impact of the delay on overall project duration.

**Table 10.** High-level summary of results for update period 1 of sensitivity analysis.

| Period | Impact on | | Project Duration | Corrective Action |
| --- | --- | --- | --- | --- |
| | Productivity [1] | Total Project Duration | | |
| Baseline | - | - | 246 days ($\sigma = 4$) | - |
| 1 | ≈50% | Notable | 251 days ($\sigma = 4$) | Required |

[1] Weather-sensitive activities.

Face validation was also used to assess the advantages and applicability of the proposed method. The results of the case study and sensitivity analysis were presented and discussed with the subject matter experts described in Table 4. The experts confirmed that the results generated by the framework were reasonable and consistent with what was expected in practice. They agreed that the framework solved existing challenges with lookahead scheduling that are not currently addressed by current commercial scheduling software, and that results generated by the proposed framework can be used to enhance and further support existing decision-making. For example, in the unfavorable weather conditions of the sensitivity analysis, the delay expected by the short-term weather forecast could prompt practitioners to double the resources available by increasing the number of shifts or working weekends during the short-term lookahead period to compensate for the (almost 50%) loss in productivity.

To further enhance the benefits of the method, the experts noted that, in its current form, the framework was not easy-to-use—particularly by practitioners who may not be familiar with simulations. The development of a graphical user interface for the system was recommended by the experts to facilitate application of the framework in industry.

## 5. Discussion

To address these limitations, a combined discrete-event and continuous simulation method that allows practitioners to more effectively assess and understand the impact of short-term weather uncertainty on construction activities during lookahead scheduling was developed and applied to a case study of a real onshore wind farm project. As expected, favorable weather conditions experienced during the tested lookahead periods resulted in a negligible impact on the productivity of weather-sensitive activities (10% reduction in planned productivity; Table 8), which translated into an overall project delay of less than 1 day. The results of the case study, together with the validation experiments, demonstrated the ability of the proposed framework to address the four barriers limiting the performance of existing methods. Specifically, the proposed framework was shown to be capable of: (1) considering additional weather parameters, (2) considering all construction activities and their criticality, (3) integrating short-term weather forecast data, and (4) integrating as-built and progress information into the lookahead scheduling process. The results of the sensitivity analysis, which demonstrated a 50% reduction in productivity (Table 10) as a result of poor weather conditions, confirmed the responsiveness of the proposed framework.

This study has advanced the state-of-the-art by addressing four key research gaps, which have limited the application of existing methods to lookahead scheduling in wind farm construction. First, the proposed method is capable of considering the impact of three weather parameters (i.e., wind, precipitation, and temperature) on onshore wind farm projects. This is in contrast to the work of Atef et al. and Guo et al., which were limited to wind speed [2,3]. Second, the proposed simulation model is capable of modeling all construction activities of an onshore wind project. Conversely, the models designed by Atef et al. and Guo et al. remained limited to wind turbine construction [2,3]. Third, the simulation model uses an innovative combined discrete-event simulation and continuous simulation approach to facilitate modeling of both non-sensitive and weather-sensitive activities of onshore wind projects. While a combined discrete-event/continuous simulation approach was used to model a variety of construction operations, such as pipeline construction [15,60], tunneling construction [23], and building construction [61], this study represents the first application of this approach to model weather-sensitive construction activities in onshore wind projects. Fourth, the proposed framework allows the integration of both short-term weather forecasts and as-built activity durations to enable decision-support at a granular level. This is in contrast to previous studies by Guo et al. and Zhou et al., which focused on the development of wind farm construction scheduling at a master scheduling level [2,3]. Importantly, while the proposed simulation model was developed for wind farm construction operations, the methodological approach used to develop the simulation model (described in Section 3.2.1) can be applied to other project types.

Proactive scheduling approaches for offshore wind farms were also explored. Kerkhove and Vanhoucke [62] proposed a mathematical optimization model for proactive scheduling of offshore wind projects subject to weather conditions. Similar to previous studies summarized in Table 1, the model proposed by Kerkhove and Vanhoucke [62] made use of a Markovian weather generator model that relied on historical data of weather parameters, focused only on the planning phase of offshore wind farm projects, and considered only two weather parameters (i.e., wave height and average wind speed).

A comparison of the proposed framework with previous models developed to assess the impact of weather conditions on the productivity of different types of construction projects is summarized in Table 11.

**Table 11.** Comparison of proposed framework with previous studies.

| Item | Research Study | | | | | | | | | | | | | | | | |
|---|---|---|---|---|---|---|---|---|---|---|---|---|---|---|---|---|---|
| | [17] | [13] | [2] | [38] | [14] | [16] | [24] | [39] | [40] | [3] | [23] | [41] | [29] | [25] | [15] | [42] | Current Study |
| Reliance on historical weather data | ✔ | ✔ | ✔ | ✔ | ✔ | ✔ | ✔ | ✔ | ✔ | ✔ | ✔ | ✔ | ✔ | ✔ | ✔ | ✔ | x |
| Flexibility of the method to analyze additional weather parameters during execution | x | x | x | x | x | x | x | x | x | x | x | x | x | x | x | x | ✔ |
| Consideration of as-built and progress information | x | x | x | x | x | x | x | x | x | x | x | x | x | x | x | x | ✔ |
| Consideration of short-term weather forecasts | x | x | x | x | x | x | x | x | x | x | x | x | x | x | x | x | ✔ |

*5.1. Practical and Managerial Implications*

While methods designed to consider the impact of historical weather data during the planning stages of construction have been developed (as detailed in Table 1), the impact of short-term fluctuations in weather conditions on productivity were not addressed in previous studies. Indeed, current commercial scheduling software, such as Primavera and Microsoft Project [59], lack the capability to consider the impact of short-term weather on productivity. Consequently, construction companies often use an intuitive, subjective approach to consider the impact of weather during lookahead scheduling, which often results in the development of unrealistic lookahead schedules and the inability to identify and implement timely corrective actions to mitigate potential weather-related delays.

This study aimed to improve lookahead scheduling practices through a simulation-based approach that is capable of considering short-term weather information along with as-built information. This study demonstrated the practicality and benefits of the proposed approach. Specifically, the simulation-based approach was capable of generating a variety of results that can be used to support decision-making in practice by:

(1) Obtaining the expected productivity (Figures 7a and 8a) and duration (Figures 7b and 8b) of weather-sensitive activities based on short-term weather forecasts, thereby increasing the representativeness of lookahead schedules over existing methods. With a more representative prediction of activity durations, practitioners are able to allocate resources (e.g., labor, material, and equipment) to activities that may be experiencing unexpected delays in productivity. For example, if a simulated activity duration is delayed by 4 days due to unfavorable weather, the project team may choose to proactively extend working days to include weekends during the lookahead period. Or, if the weather is forecasted to cause work stoppages during the second week of the lookahead period, practitioners may choose to proactively double the number of shifts during the first week when weather conditions are expected to be favorable. Targeted actions such as these not only keep the project on schedule, but may also prevent irreversible delays that can lead to disputes.

(2) Obtaining probabilistic completion times (Figures 9b and 10b) of individual activities based on short-term weather forecasts. By obtaining a probabilistic completion time, the project team is able to make more informed decisions about what types of corrective actions they can—and are interested in—pursuing. For example, if a weather-sensitive activity has a high likelihood of being delayed due to unfavorable weather, the project team may decide to postpone delivery of material for subsequent activities to avoid crowding the worksite.

(3) Obtaining a probabilistic completion duration of the entire project (Figure 11) in consideration of as-built and short-term weather forecasts. The impact of lookahead weather delays on the overall project schedule will depend on the total float of the affected activities and whether or not the activities are on the critical path. Delay of certain activities may result in a considerable delay of the overall project, while others may not affect project duration at all. The ability to easily and quickly quantify the impact of weather-related activity delays in a specific lookahead period on the overall project duration will help the project team determine the amount of mitigation effort that should be expended to resolve the delay. For example, a delay in the pouring of the concrete foundations for multiple turbines may result in the same activity-level delay as a weather-related delay in installation of the substation drainage. However, a delay in pouring concrete foundations may have a tremendous impact on subsequent activities (and material deliveries) that depend on the completion of the foundation to begin. In contrast, delays in drainage installation for the substation will not impact other activities, thereby minimally impacting overall project duration. The effort expended by the project team to mitigate each delay, therefore, will vary tremendously (Tables 8 and 10).

(4) Obtaining confidence levels for completing the project within a specific duration (Figure 11). Due to the consideration of stochastic activity durations, together with short-term weather forecast impact, outputs of the framework are stochastic and represented by a probability distribution. The probabilistic nature of the outputs provides practitioners with more insightful information, allowing the project team to base their decision on their desired level of confidence.

Using the outputs of the proposed approach, practitioners can proactively schedule their construction tasks in response to a weather-related impact on activity and project durations, thereby enhancing construction progress, reducing weather delays and related claims, and improving the likelihood of project success. While the motivation for adopting enhanced control and monitoring strategies is often the avoidance of unexpected delays and costs, the implementation of effective project control strategies can enable practitioners to capitalize on potential opportunities that may otherwise go unnoticed. Using the proposed framework, the impact of favorable weather conditions (e.g., a warmer than average lookahead period) that result in increased productivity can be easily quantified and identified. It is anticipated that the timely access of such information—made possible by the proposed framework—may allow project managers to better plan construction activities and the delivery of needed materials to capitalize on accelerated schedules, allowing for a shortening of overall project durations.

The findings of this study have highlighted several key recommendations for practitioners performing lookahead scheduling in onshore wind farm construction:

(1) Uncertainty arising from weather risk must be quantified as thoroughly and accurately as possible to maximize the likelihood of completing the project within the duration defined in the project contract.

(2) It is recommended to begin construction activities during a period characterized by favorable weather conditions to minimize the impact of weather on the productivity of activities early in the project, thereby reducing the number of subsequent activities impacted by early weather-related delays.

(3) The impact of adverse weather should be integrated with project lookahead scheduling to more accurately predict the productivity of individual activities and the entire project.

(4) Simulation-based approaches provide a better understanding and evaluation of weather impacts on individual activities and the entire project. Moreover, simulation-based approaches have the capability to consider stochastic duration of activities (as opposed to deterministic durations), allowing these systems to model other variables (in addition to weather) and allowing practitioners to choose their desired level of confidence when making decisions.

(5)     The proposed simulation-based approach allows practitioners to more quantitatively, rapidly, and easily assess the mitigation effort required to adjust the project schedule.

*5.2. Limitations*

A few limitations of this study should be considered prior to applying the proposed framework. First, triangular distributions and Dark Sky API were used because of their simplicity and for illustrative purposes. While more sophisticated methods capable of enhancing input modeling of activity durations and weather forecasting should be explored and applied, methods for improving input modeling was beyond the scope of this study. Second, because of the novelty of wind farm construction, few historical projects were available for review. While the model is intended to be universal, innovations in wind farm construction practices or organization-specific differences may exist. It is recommended that practitioners thoroughly review the WBS and if-then rules to ensure consistency with their operations and modify the WBS and if-then rules to suit their specific needs as required. Finally, while the functionality of the model was demonstrated using a real 40 MW onshore wind project, favorable weather parameters at the start of construction resulted in a negligible impact on activity and project durations, limiting the ability of the authors to compare model-derived outputs with real results under more unfavorable (i.e., extreme) conditions. Nevertheless, the sensitivity of the model to unfavorable weather conditions was demonstrated through a sensitivity analysis, where unfavorable weather conditions resulted in notable and expected activity and project-level delays.

*5.3. Future Work*

Given the limited availability of historical data in wind farm construction, research, such as the study presented here, would greatly benefit from the implementation of data collection strategies designed to quantitatively derive relationships between weather conditions and productivity. These strategies would not only improve the accuracy of the productivity-weather relationship, but would also provide the opportunity for future research in this area to be compared to historical outcomes of real projects, thereby improving validation of future models and increasing practitioner confidence. Additionally, future work should explore the integration of emerging project monitoring technologies and tools, such as sensors, GPS, and RFID, to enhance the capture of as-built data for improved accuracy of output results. Future research should also focus on modeling the effect of extreme weather events, such as lightning, and how these rare, yet intense, occurrences affect project schedules. Finally, as proposed by the subject matter experts, future work should also focus on the development of a graphical user interface that would facilitate implementation of the method by users with limited simulation knowledge. Development of a graphical user interface would also reduce the effort required to code the model for large and complex projects.

## 6. Conclusions

Adverse weather is one of the most critical and challenging schedule-related risk factors in wind farm construction due to the wind-prone locations of these projects. Weather-related delays in wind farm construction must be monitored as accurately as possible, as predictable weather conditions are not entitled to time extensions in construction contracts. Current methods, however, only account for weather during the early scheduling stages, using historical weather data to estimate the impact of weather on project duration. However, a method capable of considering variability in short-term weather forecasts for lookahead scheduling during the execution phase of wind farm construction projects had yet to be developed. In this study, a combined discrete-event and continuous simulation model was proposed with the aim of fulfilling this need by developing a practical estimation method for predicting the impact of short-term weather forecasts on activity durations and project schedules during project execution. By addressing existing research gaps, the proposed approach was able to integrate short-term weather forecasts and as-built data to

generate outputs that are logical and capable of providing much needed decision-support to practitioners. The method proposed facilitates the ability of practitioners to monitor adverse weather impacts on project durations in the time-frame required to exert effective correct actions capable of proactively mitigating weather-induced delays and subsequent related claims.

**Supplementary Materials:** The following are available online at https://www.mdpi.com/article/10.3390/su131810060/s1. Table S1: Characteristics of historical onshore wind farm projects, Table S2: Weather parameter values for update periods 1 and 2; Figure S1: Effect of temperature on construction productivity based on results from Moselhi and Kahn [26], Figure S2: Effect of precipitation on construction productivity based on results from Larsson and Rudberg [25], Figure S3: Effect of wind speed on construction productivity of crane based on results from Guo, Chen, and Chui [2], and Figure S4: Snapshot of simulation model in *Simphony.NET*.

**Author Contributions:** Conceptualization, E.M.; data curation, E.M., P.J., and A.C.; formal analysis, E.M.; funding acquisition, S.A.; methodology, E.M.; project administration, S.A.; software, E.M.; supervision, S.A.; writing—original draft, E.M.; writing—review and editing, P.J., A.C., and S.A. All authors have read and agreed to the published version of the manuscript.

**Funding:** This study was supported by Future Energy Systems research as part of the Canada First Research Excellence Fund (CFREF FES-T11-P01).

**Institutional Review Board Statement:** Not applicable.

**Informed Consent Statement:** Not applicable.

**Data Availability Statement:** All data used in the study were provided by a third party. Direct requests for these materials may be made to the provider indicated in the Acknowledgments. Models and code that support the findings of this study are available from the corresponding author upon request and with permission from the partner indicated in the Acknowledgements.

**Acknowledgments:** The authors would like to thank DNV Energy Systems Canada Inc. for providing data and support. The authors would also like to acknowledge Catherine Pretzlaw for her assistance with manuscript editing and composition.

**Conflicts of Interest:** The authors declare no conflict of interest.

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
