# Peer review of "Simulation-Based Approach for Lookahead Scheduling of Onshore Wind Projects Subject to Weather Risk"

_sustainability, doi:10.3390/su131810060_

Round 1

Reviewer 1 Report

I'm sorry to say that but I find the manuscript inconsistent.

A lookahead scheduling is defined as adjusting the base schedule based on 14 days weather forecast. However, the also total project time is analysed.

The method can't be applied for the whole project, as the more accurate weather forecast is limited to 14 days period. Otherwise is not based on 14 days forecast but on statistical, historical data (which is in contradiction to the authors' declaration).

I find it as a serious flaw.

What is presented in the article is very similar to the quantitative risk analysis presented in the "Project Management Body of Knowledge" but applied to time issues.

It can be understood if applied to the activities within 14 days period. If applied to the whole project a "lookahead scheduling" (14 days time-span) doesn't work.

The way of considering the quality of weather forecast (the probability of appearing of certain weather conditions) is not presented at all.

Applying the simulation method the Authors stopped at presenting 60-70 % range of the most probable times. In fact, the results of simulations answer how likely is to complete a certain activity or the set of activities in a given time (the probability). This is completely omitted.

There are several unclear issues or mistakes of a bit lower importance. However, as described above, in my opinion, there is a fundamental logical mistake. It forces me not to recommend publication of the manuscript.

Reviewer 2 Report

The authors adequately addressed my comments in the first review and modified the article accordingly. Moreover, the authors edited the paper for language use. As a result, the paper reads much better now and the presented concepts can be extended and replicated to other construction project types.

Reviewer 3 Report

Title: This is suggesting.

Abstract: Although is well driven, results of the method applied to the case study should be included.

Introduction: Tools of monitoring and control must be considered once more (and new) information is known after the schedule planned.

The gap is well presented.

I recommend to pay attention to delays... and advances. If the weather progresses better than expected, schedules can be improved and shortened (risks: threats and opportunities).

Literature Review: The review is well structured. However, l repeat two questions:

1. Monitoring and control tools. Applicability, weaknesses and strengths of existing tools.

2. To expand the concept of risk to threats and opportunities.

Proposed Framework: Precipitation, air temperature, and windspeed must be better justified as parameters selected. Table 1 only includes these three. Other parameters such as sky temperature, atmospheric pressure, cloudiness, wind direction, for example, are ignored without prior robust explanation.

The information that is being known is very important (for example, in bayesian propability). This information refers to activities which have began and provide new information. If only external factors are considered, the result may be poorer (I refer how actual weather conditions are affecting to construction activities once these have been started).

The idea of introduce DES in the schedule of construction activities is interesting.

Stopping criteria are ver important in real world. I think this needs further development and discussion. More references aligned would be welcome.

Case Study: Figure 6 should have a better resolution. Size of Figure 7-11, 13 are not enough (as done in 12, 14-15).

Improvement achieved thanks to the method proposed should be emphasized. A Table that compiles major findings would be welcome as well.

Discussion: The discussion is the section in which authors must delve into the meaning, importance and relevance of their results. It should focus on explaining and evaluating what you found, showing how it relates to their literature review and research questions, and making an argument in support of their overall conclusions. There are many different ways to write this section, but they can focus their discussion around four key elements:

  • Interpretations: what do the results mean?
  • Implications: why do the results matter?
  • Limitations: what can’t the results tell us?
  • Recommendations: what practical actions or scientific studies should follow?

  Conclusions: Limitations and future research should be included.

Reviewer 4 Report

1. Please comparatively evaluate your study with the articles given below, as they are very similar in subject.

Zhou et al. (2021) Stochastic resource-constrained project scheduling problem with time varying weather conditions and an improved estimation of distribution algorithm. Computers and Industrial Engineering, 157, 107322.

Kerkhove and Vanhoucke (2017) Optimised scheduling for weather sensitive offshore construction projects. Omega, 66, 58-78.

2. Please discuss your findings and results with regard to similar previous papers in the related literature. Your current section for discussion is far from a detailed comparative discussion, indeed.

3. In the section for future work, you give research implications. However, throughout the manuscript, I could not see practical and social implications. Please add a separate section for them as well.

Round 2

Reviewer 1 Report

I'm sorry to say that, but my previous remarks are still valid.

The method can't be applied to the whole project. If it is applied to the whole project (or to the remaining part of the project, longer than 14-days) the advantage of a precise short-term weather forecast disappears.

If it is applied to a 14-day period, the results are valid for that period only. The total delay can't be assessed (better than without the proposed method) as weather forecasts are precise for a 14-day period.

If the weather forecast is precise for a 14-day period, why apply simulation to calculate possible duration of a 14-day subset of the scheduled tasks. For the known weather and known lowered productivity factors the duration can be calculated (it doest have to be simulated).

In fact, the widely known PERT method is applied without mentioning that.

My experience (17 yrs in the construction industry, then 13 yrs at a university) shows that for the remote (greenfield projects as road construction, wind farms)  short-term, weather-sensitive planning is crucial and it's really applied in the construction industry. If a weather-sensitive task can't be executed, other tasks are the subject of activity. The reality is just like that.

The authors claim the novelty of the approach. It is questionable. As well as PERT technique, simulations are widely described for construction projects (the wind farm construction project is one of them).

There can be found numerous articles about managing the construction contract (with limited resources and changing time schedule). The works of prof. Piotr Jaśkowski and prof. Sławomir Biruk (Lublin University of Technology, Poland) can be checked as an example. The wind farm is still a construction project.

The limitations described at the end of the introduction as not described in the literature or not practicioned are not true.

The wind speed influence on a construction project is described in more than 3 articles (see lines 192-193).

Other (than wind) weather conditions have been already analysed too. (e.g. in 10.1016/S1644-9665(12)60123-X)

As I understand the proposal of the authors is to optimize the usage of resources within 14-days period (based on precise wether forecast and lowered productivity for different tasks). Doing that without relation to the resources being in the disposal it is not possible.

Reviewer 4 Report

May be accepted as is.